# Finding the best fit: examining the decision-making of augmentative and alternative communication professionals in the UK using a discrete choice experiment

Edward J D Webb [1,2] Yvonne Lynch [3] David Meads,[1,2] Simon Judge [3,4] Nicola Randall,[3,4] Juliet Goldbart,[3] Stuart Meredith,[3] Liz Moulam [3] Stephane Hess,[2,5] Janice Murray[3]

[1]Leeds Institute of Health Sciences, University of Leeds, Leeds, UK
[2]Choice Modelling Centre, University of Leeds, Leeds, UK
[3]Faculty of Health, Psychology and Social Care, Manchester Metropolitan University, Manchester, UK
[4]Barnsley Assistive Technology Team, Barnsley Hospital NHS Foundation Trust, Barnsley, UK
[5]Institute of Transport Studies, University of Leeds, Leeds, UK

**Correspondence to**
Dr Edward J D Webb;
e.j.d.webb@leeds.ac.uk

## ABSTRACT

**Objectives** Many children with varied disabilities, for example, cerebral palsy, autism, can benefit from augmentative and alternative communication (AAC) systems. However, little is known about professionals' decision-making when recommending symbol based AAC systems for children. This study examines AAC professionals' preferences for attributes of AAC systems and how they interact with child characteristics.

**Design** AAC professionals answered a discrete choice experiment survey with AAC system and child-related attributes, where participants chose an AAC system for a child vignette.

**Setting** The survey was administered online in the UK.

**Participants** 155 UK-based AAC professionals were recruited between 20 October 2017 and 4 March 2018.

**Outcomes** The study outcomes were the preferences of AAC professionals' as quantified using a mixed logit model, with model selection performed using a step-wise procedure and the Bayesian Information Criterion.

**Results** Significant differences were observed in preferences for AAC system attributes, and large interactions were seen between child attributes included in the child vignettes, for example, participants made more ambitious choices for children who were motivated to communicate using AAC, and predicted to progress in skills and abilities. These characteristics were perceived as relatively more important than language ability and previous AAC experience.

**Conclusions** AAC professionals make trade-offs between attributes of AAC systems, and these trade-offs change depending on the characteristics of the child for whom the system is being provided.

## Strengths and limitations of this study

► This was the first discrete choice experiment (DCE), and only the second stated preference study in the field of augmentative and alternative communication (AAC).

► The study used unusual and innovative methodology by (1) using a Best-worst Scaling case 1 study in attribute selection, (2) having AAC system choices be made in the context of a child vignette formed from a set of attributes and (3) introducing a new measure termed relative interaction attribute importance to interpret results.

► Child vignettes were relatively simple, and a single vignette could represent children with very different needs.

► In some ways, the DCE task differed from how augmentative and alternative professionals make decisions in practice.

## INTRODUCTION

Many people lack the ability to produce intelligible speech to meet their functional needs for a wide range of reasons, including cerebral palsy, intellectual/developmental delays and autism spectrum disorder. Even within disability types, individuals' communication-related needs and abilities are extremely varied. Augmentative and alternative communication (AAC) refers to methods of supporting communication. AAC systems encompass unaided methods including signing, facial expressions, body language and the use of aided systems.[1] This article focused on aided systems, also known as communication aids, which include high-tech electronic devices, such as those used by Stephen Hawking or Britain's Got Talent winner Lee Ridley, as well as low-tech systems, such as boards and communication books.

AAC can improve the lives of people with communication disabilities.[2–4] Appropriate AAC is especially important for the estimated 1 in 200 children in the UK[5–7] who require these kind of supports. Not only are their language and communication abilities still

developing and their needs evolving,[8–10] the systems used in childhood can potentially have impacts lasting a whole lifetime.[4]

Major advances in the AAC landscape have occurred in recent years.[11 12] These include technological innovation, for example, iPads and eye-tracking, though low-tech systems may still offer the best solution in many cases.[13 14] Another development within services is a greater expectation of participation in all aspects of life for people who use AAC,[11 15–17] coupled with advocacy for the right to communicate.[14] New possibilities for AAC have been created by new communication methods such as text messaging,[18] email[19] and social media.[20 21]

Despite the benefits AAC can offer, high rates of abandonment (30%–50%) of AAC systems by children have been observed,[22 23] with causes of abandonment not well understood. AAC systems can be costly (up to £10 000) and require a large amount of professional support.[24] However, when recommended appropriately and well implemented, AAC systems have been suggested to be a cost-effective use of the UK's National Health Service resources.[25]

The process through which children receive AAC systems varies, both across and within countries.[26–28] In the UK, the context for this study, children's needs and abilities are commonly assessed by a team of AAC professionals which may include speech and language therapists, occupational therapists and/or specialist teachers.[29 30] Final recommendations and decision-making about AAC systems are made with variable input from the child and family.

Choosing an AAC system requires consideration of many features. For example, what type of graphical symbols (eg, photos/stylised pictures/words) to use, how many symbols are available, how they are organised and how they are accessed.[10 31 32] The large degree of heterogeneity in the population of people who benefit from AAC, and in the systems available, means the assessment and subsequent matching of individual and system is a complex task and unique to each person.[26 28 33]

There is currently a lack of documented evidence for assessment and decision-making processes,[33–35] and what does exist is largely individual case studies.[3 25 36] AAC professionals must often make difficult and complex decisions in a complicated, heterogeneous and rapidly evolving environment, balancing the needs of an individual child and available resources.[30 37] They must also take account of the cultural and contextual influences shaping each assessment.[13 38] While there have been studies which have highlighted some important factors in decision-making,[6 33 39] available guidelines have tended to focus on the organisational structure of services, rather than decision-making as such.[29 34 40]

The current study addressed the knowledge gaps by providing quantitative evidence about AAC professionals' decision-making using a survey method termed a discrete choice experiment (DCE). DCEs are commonly used in healthcare,[41–43] and can quantify the preferences of patients, health professionals and the public for treatments, service delivery methods, policies or other things. In this case, the goal was measuring the preferences of AAC professionals when choosing AAC systems.

This study was part of a wider project entitled *Identifying Appropriate Symbol Communication aids for children who are non-speaking: enhancing clinical decision making* (I-ASC) which examined provision of AAC systems for children in the UK. I-ASC had several components, using different research methods[30 37 44 45] to generate a body of evidence on current practice and recommendations for best practice. This has resulted in resources to aid AAC decision-making (available at https://iasc.mmu.ac.uk).

Although there is a lack of robust evidence surrounding the decision-making process, some factors in successful adoption of AAC have been identified. An AAC system is more likely to be adopted by a motivated child[22] with good support from the child's network.[27 33 44] The AAC system must also meet a child's individual needs and circumstances which will be unique to every child.[14 22 46]

A previous study from the current research project investigated the AAC decision-making process using a Best-worst Scaling (BWS) case 1 survey.[45] This method was chosen as it could quantify which of several child and AAC system-related factors (37 in total) AAC professionals considered most and least important in decision-making.

The current study sought to complement the previous work by examining fewer factors in more detail using a DCE.[43] It aimed to quantify the clinical judgements and trade-offs AAC professionals make between different attributes of AAC systems, and how those trade-offs change depending on children's characteristics, things not possible using BWS case 1. This is the first DCE carried out in AAC, and there were challenges associated with performing a DCE with a target population of AAC professionals (for details see discussion). Thus, an additional goal was to establish the feasibility of using DCEs as a research tool in AAC.

## METHODS

### Survey development

No stated preference work existed in AAC prior to the current project, and there were a large number of potential attributes with little evidence as to which to include in a DCE. A BWS case 1 study was hence performed initially and the results used to guide attribute selection for the DCE. In line with good practice and to ensure attributes were meaningful and relevant, qualitative methods were used to generate attributes.[47 48] Attributes for the BWS study were generated through focus groups and interviews with AAC professionals, people who use AAC, their families and other stakeholders; systematic literature reviews; and input from an expert panel. For more details, see the methods section of Webb *et al*[45]

The BWS study produced relative importance scores for the 19 child and 18 system attributes given in online supplementary appendix A. DCE attributes were selected

**Table 1** Child attributes and levels including brief descriptions

| Child attributes and levels | Description* |
|---|---|
| **Receptive and expressive language (1)** | Child's ability without AAC to understand communication from others (receptive) and communicate with others (expressive). |
| Delayed† | Both receptive and expressive abilities below expectation given child's age. |
| Receptive language exceeding expressive language | Ability to understand communication from others greater than ability to communicate with others. |
| **Communication ability with AAC (3)** | How well a child can communicate when using AAC. |
| No previous AAC experience† | Has never communicated using AAC before. |
| Able to use AAC for a few communicative functions | Can use AAC for some basic functions (eg, simple requests). |
| Able to use AAC for a range of communicative functions | Can use AAC for more complex tasks, for example, constructing sentences. |
| **Child's determination and persistence (4)** | Attitude of child towards communication and using AAC. |
| Does not appear motivated to communicate through any methods and means† | Child is not inclined to develop communication skills. |
| Motivated to communicate through symbol communication systems | Child has demonstrated motivation and willingness to use AAC. |
| Only motivated to communicate through methods other than symbol communication | Child may be motivated to communicate, but is not inclined to use AAC. |
| **Predicted future skills and abilities (6)** | Professional assessment of how child's communication abilities will develop. |
| Regression† | Abilities projected to become worse in future (eg, due to a degenerative condition such as Rett syndrome). |
| Plateau | Abilities will not change significantly in future (eg, a child aged 16–17). |
| Progression | Communication abilities will develop in future. |

*Descriptions are not intended as rigorous definitions of AAC terminology, but as a rough guide for the non-AAC specialist reader.
†Indicates baseline level; numbers in parentheses indicate attributes' rank in relative importance from Webb et al.[45]
AAC, augmentative and alternative communication.

from these during consensus discussions between authors with expertise in AAC, speech and language therapy, and health economics. The selection criteria were that attributes should: (1) form coherent and realistic descriptions of children and systems, (2) address the research aims of the wider research project (eg, a focus on symbol communication systems), (3) include mainly attributes with high relative importance scores in the BWS study and (4) be small in number so choice tasks would not overburden respondents. Consensus was achieved via unstructured discussions until all authors were in agreement. This resulted in four child and five system attributes. The attributes are listed in tables 1 and 2, together with non-specialist descriptions for the benefit of the general reader. For a further introduction, see Beukelman and Mirenda.[17]

In summary, the child attributes captured a child's language ability, experience with AAC, attitude/motivation to communicate with AAC and whether the child is expected to regress, plateau or progress in communication ability. A total of 54 child vignettes were formed from the set of child attributes. Authors with expertise in AAC and speech and language therapy identified and removed 18 child vignettes representing unrealistic combinations, leaving 36.

AAC system attributes broadly captured the vocabulary set(s) provided by manufacturers, vocabulary size and organisation, type of graphical symbols used and how consistent the navigational layout of words/symbols is when accessing items. It was not stated whether a system was high-tech or low-tech, although certain levels, for example, vocabulary sets with staged progression, are more common with high-tech systems. Authors with experience in AAC and speech and language therapy removed 158 unrealistic combinations from the 432 AAC systems which could be formed from the system attributes, leaving 274.

Prior experience from the BWS study suggested it would be difficult to recruit a large respondent sample, so to maximise the information captured a relatively heavy response burden of 12 choices between three systems was selected for the DCE. Participants were shown three child vignettes, referred to as child A, child B and child C, and made four choices for each child vignette. An example task is shown in online supplementary appendix B.

The survey's statistical design (ie, which levels of the AAC system attributes were presented in each question) was generated using NGene (©ChoiceMetrics), with 60 choice tasks split into five blocks. The design sought to maximise D-efficiency, a measure of how

**Table 2** AAC system attributes and levels, including brief descriptions

| AAC system attributes and levels | Description* |
|---|---|
| **Vocabulary sets (1)** | Words and/or symbols preprovided with system 'out of the box' (eg, as part of a software package for a high-tech system). |
| No vocabulary set† | AAC practitioners/child's support network provides all vocabulary content. |
| Fixed vocabulary set | A single fixed set of vocabulary which may be customised. |
| Vocabulary set with staged progression | A series of vocabulary sets with predetermined progression through them that simulate language development. For example, an initial set including just basic words, with subsequent sets introducing more grammatical structure. May be customised. |
| **Consistency of layout (2)** | How consistent positions of words/symbols are in system interface, and how consistent navigation to find different symbols is? |
| Consistency of some aspects of layout† | Words/symbols in multiple categories appear in different positions across categories, but always in the same place in a given category. |
| Consistency of all aspects of layout | All/nearly all words/symbols always appear in same position in interface. |
| Idiosyncratic layout | Layout that has been personalised for an individual child. |
| **Type of vocabulary organisation (5)** | How words/symbols are organised within the system. |
| Visual scene† | Interface shows photos, most likely of scenes familiar to the child, with areas of it highlighted to represent words. |
| Taxonomic | Words/symbols organised according to subject, analogous to non-fiction books in a library. |
| Semantic-syntactic | Words/symbols organised according to sentence structure, for example, verbs, nouns, adjectives. |
| Pragmatic | Words/symbols organised around function in language rather than grammar, for example, request, mood. |
| **Size of vocabulary (7)** | How many words/symbols system can output. |
| Up to 50 vocabulary items† | Implies only simple communication functions possible. |
| 50–1000 vocabulary items | Implies combining words/symbols to create grammatical structures. |
| More than 1000 vocabulary items | Does not imply more complex communication than 50–1000 items, but means a greater load on child's memory. |
| **Graphical representation (12)** | Type of symbols used by system. |
| Photos† | Photographs, possibly of items or environments personal to the child. |
| Pictographic symbol set | Non-photorealist pictures with specific meanings attached. May be accompanied by text. |
| Ideographic symbol system (with rules or encoding) | Stylised symbols combined with fixed rules and grammar analogous to Chinese/Japanese characters (eg, Minspeak). |
| Text | Text unaccompanied by other symbols |

*Descriptions are not intended as rigorous definitions of AAC terminology, but as a rough guide for the non-AAC specialist reader.
†Indicates baseline level; numbers in parentheses indicate attributes' rank in relative importance from Webb *et al.*[45]
AAC, augmentative and alternative communication.

much information it is possible to extract from survey responses.[49]

The survey was piloted by five AAC professionals and consequently the wording of some attributes and levels altered.

### Survey administration

The DCE was administered online for ease of recruitment. Recruitment was carried out via AAC professionals' email distribution lists (the project's own list and the mailing list of the UK wide charity Communication Matters; www.communicationmatters.org.uk). In addition, invitations were sent via publicly available lists and websites, and the professional contacts of authors. Adverts were also placed on the project website and online media. Responses were collected between 20 October 2017 and 4 March 2018. Informed consent was obtained from participants at the start of the survey.

Participants began by confirming they contributed towards AAC decision-making for children, and those who indicated they did not progressed directly to demographic questions that were at the end of the survey (for details, see online supplementary appendix A) (The precise wording of the question was: "I confirm my work involves assessing children for aided AAC systems and I contribute to the

decision making in relation to the language and vocabulary organisation within AAC systems." During testing it was revealed that some AAC professionals did not have sufficient input into the decision-making process in their day-to-day practice for the DCE questions to be meaningful, for example, occupational therapists specialising in optimising physical access to an AAC system recommended by other members of the team, and this question was designed to filter out such respondents.) Three child vignettes and one survey block were randomly allocated to each participant. The order of system attributes was randomised between participants, but consistent within participants, and which systems appeared on the left, middle and right of the screen was also randomised.

## Analysis

Analysis of participants' choices was grounded in random utility theory. This standard approach[50] assumed participants chose the object which maximised their utility. The utility of an object was modelled as depending partly on the object's attributes and partly random, the latter component capturing the influence of all factors not included in the model. In a given choice scenario $t$, participant $i$ chose which of three AAC systems to allocate to child $c$. The utility to participant $i$ of allocating AAC system $s \in \{1, 2, 3\}$ to child $c$ in choice scenario $t$ was:

$$u_{isc} = \alpha_s + \beta_{ic}x_s + \varepsilon_i$$

where $\alpha_s$ was an alternative specific constant for AAC system $s$, $x_s$ was a vector of dummy variables indicating AAC system levels, $\beta_{ic}$ was a vector of coefficients which differ across participants and children, and $\varepsilon_i$ was a random error term.

The coefficient on level $l$ of system attribute $a$, $\beta_{ialc}$, depended on the characteristics of the child vignette according to:

$$\beta_{ialc} = \gamma_{ial0} + \gamma_{ial}z_c$$

where $\gamma_{ial0}$ was a constant giving the preference for a system attribute at baseline child levels, $z_c$ was a vector of dummy variables indicating vignette levels and $\gamma_{ial}$ was a vector of coefficients, allowing for heterogeneity in relative preference for AAC system attributes depending on child characteristics.

A full model with all interaction terms included too many parameters to estimate reliably. Thus, parameters were eliminated in a step-wise process and a final preferred mixed logit model was identified using the Bayesian Information Criterion. The mixed logit model incorporated participant heterogeneity by allowing AAC system attribute parameters to be random, following a normal distribution with both means and variances depending on child characteristics. For details, see online supplementary appendix C.

Models were estimated using the CMC Choice Modelling Centre Code for R V.1.1,[51] and all analyses were carried out using R V.3.3.1. Statistical significance was assessed at the 5% level after adjusting for multiple testing using Holm's sequential Bonferroni correction.[52]

Results were presented using a new measure termed *relative interaction attribute importance* (RIAI) which assessed how big an impact child attributes have on AAC professionals' decision-making. RIAI is analogous to relative attribute importance, often used to present DCE results,[53] and may be calculated either with respect to a single choice object attribute or overall with respect to all choice object attributes. For a formal definition of RIAI, see online supplementary appendix D.

### Patient and public involvement

One author (SM) is an AAC user, and one (LM) is the parent of an AAC user, and both were involved in all stages of research development and delivery.

## RESULTS

A total of 172 participants completed the survey, of whom 155 indicated they contributed to decision-making regarding AAC systems and answered DCE questions. Summary statistics of their demographics and professional experience are given in table 3. Most participants were female (~90%) and white. We believe this to be reasonably representative of the population of AAC professionals in the UK (eg, data from the Health and Care Professionals Council showed speech and language therapists in the UK were 96% female and the Higher Education Statistics Agency found speech and language therapy students in 2017/2018 were 79% white. Source: Royal College of Speech and Language Therapists, personal communication). The mean age of DCE participants was around 40, with a range from 24 to 65, and they had on average 10 years' experience of AAC. Around 75% of DCE participants had a speech and language therapy background, with no other background reported by more than 10%. Those who did not answer DCE questions were less likely to have a speech and language therapy background (~50%), with teacher (~20%) and occupational therapist (~30%) more common.

Approximately 30% of the sample worked with all age groups, while 50%–60% worked with preschool, primary school and secondary school aged children. Participants were asked for the three most common diagnoses encountered in their work, with ~80% stating physical disability, ~70% stating intellectual disability/developmental delay and ~65% stating autism spectrum disorder.

Turning to DCE responses, respondents chose the left-hand option 37.6% of the time, and the central and right-hand options 33.1% and 29.2% of the time, respectively, significantly different from an equal distribution (one sample Kolmogorov-Smirnov p=0.002).

Table 4 contains the results of the final preferred model, with 24 coefficients. Figure 1 illustrates the RIAI of child attributes for each system attribute and overall. The constant terms in table 4 give participants' preferences for AAC system allocation when shown a child vignette

**Table 3** Demographics and professional experience of participants

| | Mean | SE |
|---|---|---|
| Age (years) | 40.8 | 11 |
| Experience (years) | 11.4 | 9.2 |
| % of role relating to AAC | 53.7 | 34.3 |
| | **N** | **%** |
| Gender | | |
| Female | 155 | 90.1 |
| Male | 10 | 5.81 |
| Prefer not to say | 7 | 4.07 |
| Ethnicity | | |
| White (English/Welsh/Scottish/Northern Irish/ British) | 149 | 86.6 |
| White (other) | 12 | 6.98 |
| Other | 6 | 3.49 |
| White (Irish) | 5 | 2.91 |
| Professional background | | |
| Speech and language therapist | 125 | 72.7 |
| Occupational therapist | 16 | 9.3 |
| Teacher | 14 | 8.14 |
| Other | 12 | 6.98 |
| Assistive technology specialist | 5 | 2.91 |
| Clinical scientist | 5 | 2.91 |
| Age groups worked with | | |
| Primary school age | 99 | 57.6 |
| Secondary school age | 94 | 54.7 |
| Preschool age | 85 | 49.4 |
| All age groups | 56 | 32.6 |
| Higher education | 30 | 17.4 |
| Further education | 21 | 12.2 |
| Other | 12 | 6.98 |
| Adults | 10 | 5.81 |
| Among most common three diagnoses seen in practice | | |
| Physical disability (eg, neuromuscular, cerebral palsy, etc) | 140 | 81.4 |
| Intellectual Disability/Developmental Delay | 118 | 68.6 |
| Autism spectrum disorder | 113 | 65.7 |
| Syndromes | 61 | 35.5 |
| Neurological | 45 | 26.2 |
| Specific Speech/Language Impairment | 22 | 12.8 |
| Dyspraxia | 14 | 8.14 |

For some questions, participants could select more than one response, thus some percentages do not sum to 100%.
AAC, augmentative and alternative communication.

with all attributes at baseline levels. This baseline vignette is as follows: "Child A/B/C has delayed expressive and receptive language and no previous AAC experience. Child A/B/C does not appear motivated to communicate through any methods and means. Child A/B/C is predicted to regress in skills and abilities (regression)." It represents what was considered by the researchers as the most challenging profile that can be formed from the set of child attributes.

The interaction terms represent how respondents' preferences for AAC systems changed if choosing for a child vignette which differed on a given child attribute.

### Vocabulary sets
For the baseline child vignette, vocabulary sets which are fixed or have staged progression were preferred to no preinstalled vocabulary. Only a single child attribute influenced preferences: Professionals were much more likely compared with the baseline to choose systems with staged progression vocabulary sets over no preinstalled set if the child vignette was predicted progress in skills and ability (odds ratio (OR) 3.88) (table 4).

### Consistency of layout
For the baseline child vignette, a consistent layout or an idiosyncratic layout was preferred to only having some aspects of system layout consistent for use, with no interactions with child attributes (table 4).

### Vocabulary organisation
For the baseline child vignette there was no significant preference between visual scene, taxonomic or semantic-syntactic vocabulary organisation, while pragmatic organisation was preferred. There were two significant interactions between vocabulary organisation and motivation. A child vignette with motivation to communicate using AAC became more likely to be allocated a system with taxonomic (OR 2.03) or semantic-syntactic (OR 2.29) organisation compared with visual scene layout (table 4).

### Size of vocabulary
For the baseline child vignette there were no significant differences in preferences between up to 50 and between 50 and 1000 vocabulary items, but over 1000 items were considered significantly less appropriate. A mid-size vocabulary (50–1000 items) became more preferable compared with 50 or fewer for a child vignette motivated to communicate using AAC. Over 1000 items became significantly more preferable for child vignettes with each of the following characteristics: Receptive language exceeding expressive language, an ability to use a range of AAC functions, motivated to communicate using AAC and predicted to progress (table 4). All child attributes influenced preferences for vocabulary size. As measured using RIAI, communication ability with AAC (32%) and determination and persistence (28%) were relatively more important than future skills and abilities (22%) and receptive and expressive language (17%) (figure 1).

### Graphical representation
For the baseline child vignette there was no preference between graphical representation using photos or pictographs, but text was less preferred than either, and

**Table 4** Parameter means and SD for final mixed logit model

| AAC system attribute | Child attribute | | Parameter mean | SE | σ | SE |
|---|---|---|---|---|---|---|
| Vocabulary sets (baseline none) | Fixed | Constant | 0.283* | 0.0966 | 0.131 | 0.258 |
| | Staged progression | Constant | 0.364* | 0.141 | 0.941* | 0.206 |
| | | Predicted to progress | 1.36* | 0.221 | −1.09* | 0.343 |
| Consistency of layout (baseline some aspects) | Consistency of all aspects | Constant | 0.892* | 0.121 | 0.15 | 0.126 |
| | Idiosyncratic layout | Constant | 1.46* | 0.14 | 0.757* | 0.134 |
| Type of vocabulary organisation (baseline visual scene) | Taxonomic | Constant | 0.0629 | 0.165 | 0.383 | 0.257 |
| | | Motivated to communicate through symbol communication systems | 0.707* | 0.206 | −0.563 | 0.295 |
| | Semantic-syntactic | Constant | −0.178 | 0.166 | 0.549 | 0.234 |
| | | Motivated to communicate through symbol communication systems | 0.826* | 0.197 | −0.112 | 0.296 |
| | Pragmatic | Constant | 0.443* | 0.123 | 0.723* | 0.152 |
| Size of vocabulary (baseline 50 items) | 50–1000 items | Constant | 0.131 | 0.143 | 0.43 | 0.166 |
| | | Motivated to communicate through symbol communication systems | 1.01* | 0.232 | −0.731 | 0.329 |
| | More than 1000 items | Constant | −0.929* | 0.213 | 1.02* | 0.33 |
| | | Receptive language exceeding expressive language | 0.692* | 0.186 | 0.489 | 0.367 |
| | | Able to use AAC for a range of communicative functions | 1.14* | 0.319 | −0.419 | 0.762 |
| | | Motivated to communicate through symbol communication systems | 1.31* | 0.272 | −0.751 | 0.556 |
| | | Predicted to progress | 0.902* | 0.233 | 0.981 | 0.657 |
| Graphical representation (baseline photos) | Pictographic symbol set | Constant | −0.41 | 0.183 | 0.0722 | 0.248 |
| | | Motivated to communicate through symbol communication systems | 1.36* | 0.24 | −0.363 | 0.428 |
| | | Predicted to progress | −0.814* | 0.217 | 1.12 | 0.385 |
| | Ideographic symbol system | Constant | −1.25* | 0.207 | 0.823* | 0.216 |
| | | Motivated to communicate through symbol communication systems | 1.67* | 0.268 | 0.069 | 0.297 |
| | Text | Constant | −0.709* | 0.159 | 0.615* | 0.204 |
| | | Motivated to communicate through symbol communication systems | 1.39* | 0.231 | −1.12* | 0.282 |

Constants give preferences when choosing for the baseline child vignette: 'Child A/B/C has delayed expressive and receptive language and no previous AAC experience. Child A/B/C does not appear motivated to communicate through any methods and means. Child A/B/C is predicted to regress in skills and abilities (regression).' σ indicates SD. Parameter variance for level of *l* AAC system attribute *a* when choosing for child is given by $\sigma^2_{alc} = \left(\sigma_{al0} + \sigma_{al}Z_c\right)^2$.
*Significance at the 5% level corrected using Holm's sequential Bonferroni.
AAC, augmentative and alternative communication.

idiographic symbols were even less preferred. Interactions were seen with two child attributes. Motivation to communicate using AAC increased the probability of choosing pictographic symbols (OR 3.88), idiographic symbols (OR 5.31) or text (OR 4.00) rather than photos. However, being predicted to progress made pictographic symbols less preferable (table 4).

### Overall RIAI of child attributes

Overall, future skills and abilities had the highest RIAI (38%), followed by child's determination and persistence (19%), communication ability with AAC (20%) and receptive and expressive language (12%) (figure 1).

### DISCUSSION

This DCE has revealed AAC professionals' priorities when choosing AAC systems for children, and shown that these priorities change when faced with children with different characteristics. That priorities change in this way is not unexpected, and in line with previous research showing that AAC professionals recognise the importance of matching an AAC system to an individual person's needs.[22 54] However, the current study builds on previous findings by showing the magnitude of preference changes, as for some system attributes their preferences for different levels could completely reverse depending

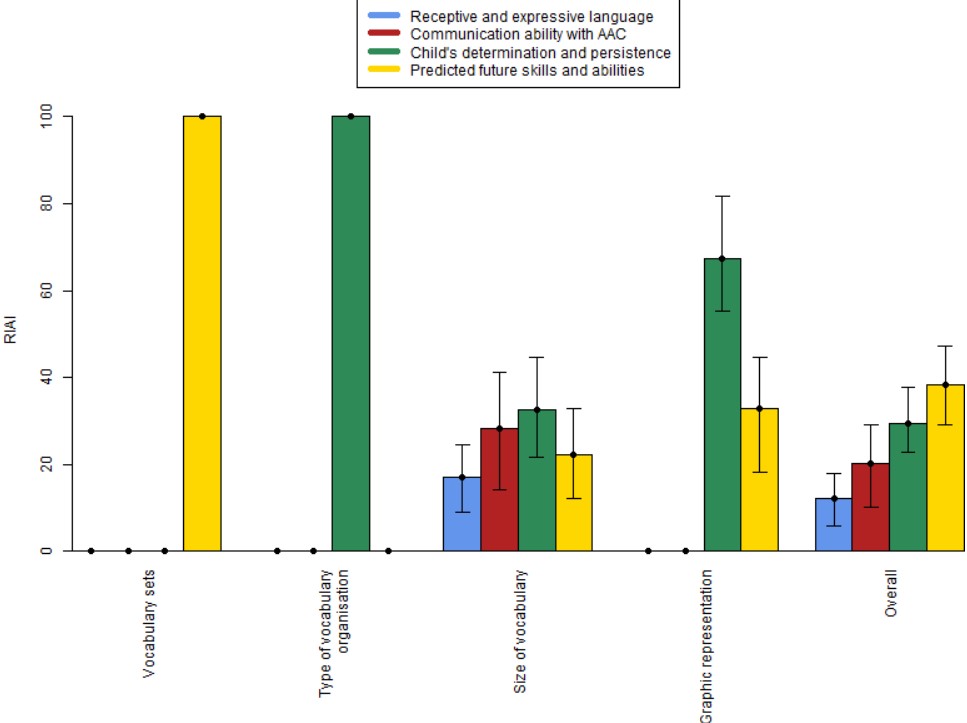

**Figure 1** Relative interaction attribute importance for each AAC system attribute and averaged over all attributes. Note that consistency of layout is omitted as there are no interactions with child attributes. Error bars show 95% CIs. AAC, augmentative and alternative communication.

which child vignette was shown. For example, for the baseline child vignette (see table 4), a system with more than 1000 vocabulary items was less likely to be chosen than one with fewer than 50 (OR 0.395). However, for a child vignette describing a receptive-expressive language gap, the ability to use AAC for a range of functions, motivation to use AAC and predicted progression, a system with more than 1000 vocabulary items was more likely to be chosen (OR 22.5). Such flexibility is encouraging, as it is in line with one of Williams et al's[14] five principles for AAC application: 'AAC systems must be highly individualised and appropriate to individual needs' (p195).

A key finding was that the attribute of the child's determination and persistence had the greatest number of interactions with preferences and was more important in terms of RIAI than language ability or previous experience with AAC. Specifically, the attribute level motivation to communicate using AAC tended to drive participants towards what can be regarded as more ambitious choices, for example, more vocabulary items. It may be that participants believed that motivated children are more likely to succeed with such AAC systems, in line with previous findings that attitude towards AAC and valuing an AAC system are important factors in successful adoption of AAC.[22 54]

Visual scene vocabulary organisation and graphical representation using photos can both involve items/ scenes from an individual's own life, and use literal, rather than abstract depictions. Both were less preferred for child vignettes motivated to communicate via AAC. Rather, participants favoured more abstract methods of

organisation (taxonomic and semantic-syntactic) and graphical symbols that require more grammar (pictographs, ideographs and text). Preferences for abstract methods of organisation and symbols requiring more grammar may be interpreted as an unfounded[55] belief that motivated children will be better able to use more complex AAC systems. An alternative and by no means mutually exclusive interpretation is that lack of motivation requires an AAC system involving familiar cues from their everyday environment.

Previous studies have also studied how AAC professionals choose graphical symbols for children.[56] For example, Thistle and Wilkinson[33] found that cognitive abilities are an important factor, as did Dada et al.[46] The advantage of a DCE was that the precise interactions between child characteristics and symbol type have been enumerated, showing, for example, which children were more likely to be given AAC systems with photos, and which were more likely to be given systems with text. AAC system preferences did not significantly differ between child vignettes where their skills and abilities were predicted to regress or plateau. However, if a child was predicted to progress, this had a large impact on professional decision-making, with anticipated future skills and abilities ranked as the highest attribute in terms of RIAI. As with motivation, skills and abilities led to more ambitious choices, with more vocabulary items preferred and pictographs depreciated compared with ideographs and text. Such ambitious choices could reflect participants wishing to provide AAC systems that would fulfil the future needs of children who are anticipated to progress,

given the large investment that goes into learning a new AAC system.[57–59] With plateau or regression this was less of a concern.

Photos were still the most preferred aided communication mode unless a child vignette featured both predicted progress *and* motivation to communicate via AAC. This preference for photos possibly indicates that photos remain a good starting point for a child who is not engaged, regardless of prognosis, and may reflect recommendations that recognise the need to reduce the learning demands of AAC systems for some children.[12 60]

Despite unwelcome rates of abandonment, AAC professionals had high expectations of motivated children who were expected to progress, even if their receptive and expressive language were both delayed and they had no previous AAC experience. It has previously been noted that people who use AAC experience an asymmetry between the language they receive and the language they are able to express.[61] One interpretation is that participants wished to minimise asymmetries by choosing text as the expressive output for children they believed could cope with it. These ambitious choices are also encouraging given the greatly increased aspirations for effective societal participation of AAC users.[11 15 16] It is also in line with official guidance[62] and one of Williams *et al*'s[14] five principles for AAC: 'AAC must support full participation in all aspects of 21st century life' (p195).

For many of the child vignettes there were non-linear preferences for vocabulary size. Offering between 50 and 1000 items was considered better than 50 or fewer for all child vignettes, although the difference was not always significant. Systems with fewer than 50 items being depreciated may indicate that participants were mindful of limiting children's potential for expression, even for children with lower cognitive ability and poor prognosis.

Findings suggest that respondents preferred levels of AAC systems that require personalisation, for example, pragmatic vocabulary organisation or an idiosyncratic layout, in line with previous findings that personalisation is important in successful AAC adoption.[28] A preference for personalisation indicates that it is not possible to achieve the goal of AAC systems being closely tailored to individuals' needs[14 63] with 'off-the-shelf' AAC systems: in other words, some personalisation is always necessary.[64 65]

Preinstalled vocabulary sets were always preferred over no preprovided set, in line with other studies showing that selecting core vocabulary was an important part of AAC professionals' decision-making process.[33 37]

Comparing the DCE results with the previous BWS case 1 study,[45] some similarities may be observed. For example, graphical representation was the lowest ranked attribute in terms of importance in the BWS to be included in the DCE. In concordance with this finding, when the relative importance of AAC system attributes was calculated for each child vignette in the DCE, graphical representation was never the most important attribute. The relative lack of importance ascribed to graphical representation raises debate about the fundamental components of language

construction through aided means and suggests much further research is required.

Many differences to the BWS findings can also be seen. Language abilities were the most important child attribute in the BWS, yet its RIAI in the DCE was below predicted future abilities, ranked sixth in the BWS. However, differences do not necessarily imply contradiction, as the two methodologies did not measure the same thing. The BWS measured the importance of AAC system attributes over the case mix AAC professionals encounter in practice, whereas for the DCE respondents were presented with a specific child vignette.

Receptive and expressive language had the lowest RIAI overall, with only a single interaction term in the final model. This contrasts with some previous findings that a child's language abilities play a large role in selecting an appropriate AAC System.[13 28 30 37] One possible explanation is that the aspects of language ability which were most relevant were captured in this study by other child attributes, but this remains a question to be addressed by future research.

The current study has demonstrated the feasibility of conducting a DCE with a target population of AAC professionals. The demonstration of feasibility is noteworthy given the relative rarity of DCEs studying health professionals' decision-making. For example in a systematic review[41] of DCEs in health published between 2013 and 2017, only 13% included a sample of health professionals. In addition, there were particular challenges associated with performing a DCE with AAC professionals. The target population in the UK is small, meaning it was uncertain that sufficient participants for a successful study could be recruited. There were also concerns that participants might not find the DCE format acceptable, as they might have rejected having to make compromises between AAC system attributes in the context of providing a system for a child. Yet despite informal feedback that some respondents found the tasks uncomfortable, many were still willing to complete them. Finally, as interactions between child characteristics and AAC systems are so important, it was necessary to present hypothetical child vignettes, making tasks more complicated than in a typical DCE.

Despite these potential pitfalls, the DCE was successfully carried out, and having demonstrated the feasibility of the method in this area, further DCE studies should be considered in future.

## Limitations

The current study has several limitations. The sample size was relatively small (155 participants, compared with a median for healthcare DCEs of 401[41]). However, many studies exist with smaller sample sizes (eg, Spinks *et al*[66] with 35), and it was possible to estimate robust statistical models. Furthermore, it would have been difficult to collect a larger sample, as 155 participants represented a large proportion of the population of AAC professionals in the UK working with children, which was estimated

at around 800 (Communication Matters, personal correspondence).

The DCE task may not match how UK AAC professionals make decisions in practice. Typically, many participants have the opportunity to work with families and children, as well as part of an AAC team, which could include diverse areas of clinical and personal expertise. Teams also generally make recommendations, rather than unilaterally choosing a system. However, there is evidence that AAC professionals compare the attributes of AAC systems in everyday practice,[13] and that they make trade-offs between system attributes,[37] akin to DCE tasks. In addition, it is still useful to study the individual decision-making of AAC professionals. Lynch et al[30] reported that a wide variety of team structures are used, and the mode of service delivery can have an influence on outcomes. Gathering evidence on individual-level decision-making can thus inform an assessment of how different ways of organising services influence decisions.

The DCE tasks presented one-off static decisions made by a single individual. In reality the decision-making environment is dynamic, with children developing over time, and often having two or more devices over the course of their childhood. These differences are a limiting factor in the external validity of results.

Attributes and levels use a mixture of speech and language therapy terms (eg, receptive and expressive language) and more AAC-specific language (eg, staged vocabulary progression). Mixing these terms may have made it more difficult for respondents from any one professional specialty to interpret all of them.[44] However, this issue is not limited to the current study, but reflects an ongoing struggle in AAC to establish a common language.[44] In addition, respondents may have been unfamiliar with the generic term ideographic symbols, since only a single commercial set of ideographic symbols is in popular use in the UK (Minspeak, Semantic Compaction Systems).

Respondents were more likely to choose AAC systems on the left of the screen and less likely to choose ones on the right. However, the risk of bias was mitigated by allowing for alternative specific constants and randomising the position in which AAC systems were presented.

Compared with the real children AAC professionals encounter, the child vignettes were simple, and lacked information which influenced decision-making, such as the child's preferences[33] and contextual factors.[30] However, this is an inherent limitation of the DCE methodology, and vignettes with a greater number of attributes and levels would have made decisions overly burdensome, and therefore were not included. Significant interactions between AAC systems and child attributes implied that the vignettes were meaningful enough that respondents changed their preferences in response to them, often dramatically.

For a given child vignette, it was only possible to determine relative preferences for system attributes, rather than absolute preferences. Consequently, it is not possible to tell how suitable a given system is for a given child vignette which is important as some presented a challenging profile, for which it may be hard to find a suitable AAC system.

## CONCLUSION

A lack of rigorous evidence on how to best assess and provide AAC systems for children has previously been identified,[25 34 44] as well as a gap between research and current practice.[11] In light of this, the current study's results are encouraging, as it shows AAC professionals following best practice in many areas, for example, ensuring AAC systems suit individual needs, and having high expectations for many children.

However, there is still demand from AAC professionals for better support in decision-making,[33 37] and undoubtedly current practice could be improved. The results of the current study, together with evidence from the wider research project, have been used to create a heuristic and suite of resources (available at https://iasc.mmu.ac.uk). It is hoped that these resources will aid AAC professionals in their clinical practice and help them provide the best possible service for children.

**Acknowledgements** Thank you to Muireann McCleary and the Speech and Language Therapy team at the Central Remedial Clinic, Dublin, who piloted and gave feedback on the survey, and to participants who responded to the survey. Thanks to Mark Jayes (Manchester Metropolitan University) and Berenice Napier (Royal College of Speech and Language Therapists) for assistance in finding demographic statistics on speech and language therapists in the UK.

**Contributors** All authors conceived the study and defined the study aims. EW, DM, YL, NR, SJ, JG, SM, LM and JM developed attributes and levels. EW, DM and SH constructed the survey statistical design. EW and DM collected data. EW conducted statistical analysis. EW, YL, NR, SJ, JG, SM, LM and JM interpreted findings. EW wrote the manuscript first draft. All authors contributed to and approved the final manuscript.

**Funding** This independent research was funded by the National Institute for Health Research, UK (Health Services & Delivery Research Project: 14/70/153 - Identifying appropriate symbol communication aids for children who are non-speaking: enhancing clinical decision-making). The views expressed in this article are those of the authors and not necessarily those of the NHS, the National Institute for Health Research or the Department of Health. Stephane Hess acknowledges additional support by the European Research Council through the consolidator grant 615596-DECISIONS.

**Competing interests** None declared.

**Patient consent for publication** Not required.

**Ethics approval** Ethical approval was received for the study from an NHS Research Ethics Committee (REC reference 6/NW/0165) and informed consent was obtained from participants at the start of the survey.

**Provenance and peer review** Not commissioned; externally peer reviewed.

**Data availability statement** Data are available upon reasonable request.

**ORCID iDs**
Edward J D Webb http://orcid.org/0000-0001-7918-839X
Yvonne Lynch https://orcid.org/0000-0003-3209-3099
Simon Judge https://orcid.org/0000-0001-5119-8094

Liz Moulam https://orcid.org/0000-0003-3810-1037

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
