## [Reviewer comments · BMJ Open]

ARTICLE DETAILS

TITLE (PROVISIONAL)	Finding the best fit: Examining the decision making of augmentative and alternative communication professionals in the UK using a discrete choice experiment
AUTHORS	Webb, Edward; Lynch, Yvonne; Meads, David; Judge, Simon; Randall, Nicola; Goldbart, Juliet; Meredith, Stuart; Moulam, Liz; Hess, Stephane; Murray, Janice

VERSION 1 – REVIEW

REVIEWER	Teresa Iacono La Trobe University Australia
REVIEW RETURNED	31-Mar-2019

GENERAL COMMENTS	This study was described as a discrete choice experiment (DCE). It provided a systematic method to quantify decisions made by professionals in recommending AAC for children, which was tested in the DCE. I appreciated the opportunity to review the manuscript (Ms), which could make a valuable contribution to both research and practice in AAC, particularly in an environment of cost rationalisation. In general, the study seemed solid, but the Ms was at times very difficult to follow, requiring some attention to organisation and writing, and to review of the literature, and to relating findings to previous research in the discussion. My concerns are detailed below. 1. In the abstract, I suggest not using abbreviations without defining them – DCE being a case in point. Also, there were many incomplete sentences, including at lines 16, 21, 27.2. The language and expression suggested a very medical model focus, such as through referring to various types of disability as “conditions” and removing reference to the child when discussing the “vignettes” – e.g., p. 12, line 7, “a vignette with a receptive-expressive language gap,” and line 32 “a vignette motivated to communicate.” Similarly, I found the expression “a type of child seldom encountered” (p. 60) to be quite jarring. I suggest re-wording as “a child with a combination of characteristics seldom encountered.” Can I also suggest that people are “patients” (p. 9) only in health care settings. AAC decision-making does not always occur in a health setting, unless this is the case in the UK? I wonder if “consumer” could be substituted.3. As this journal has a wide readership, with many unfamiliar with AAC, a number of terms require definition – such as “pre-installed” vocabulary (although, if in relation to high technology, this is more usually referred to as “pre-programmed”), “visual scene display,” “pragmatic organisation,” “fixed or staged progression” (admittedly, although familiar with AAC literature, I don’t know what is meant by the latter).
---

	4. The introduction would benefit from more information about previous relevant research. Citations are used to support statements, but this section should give the reader a sense of the state-of-the-art in relation to the topic of the study. I didn't get this sense. 5. A stronger or perhaps more complete rationale is needed for the study. I can see great value in knowing how clinicians make decisions about AAC recommendations and in having a research tool that can be applied in contexts other than the UK. The tool and future research that explored the alignment of decisions and best practice recommendations (such as those of Beukelman and Mirenda's Participation Model, or clinical guidelines available through speech-language therapy/ pathology associations across the world) would certainly help in evaluation of practices. I wonder if the rationale could be strengthened, followed by specific aims (they currently are filtered through the latter part of the introduction) that better capture the focus on children and aided AAC. On the latter point, it was unclear if this was all aided AAC or only high technology, in light of the featuring of pre-installed vocabulary. Currently, the last paragraph of the introduction ends with addressing "both these aims" – but I struggled to know exactly what these were. 6. The study is contextualised as though it would be read by a UK audience only. Consideration needs to be given to the international readership: currently assumptions about who and how decisions are made about AAC are related to the UK only. For example, AAC decision-making does not always happen in teams. Also, reference to UK policies or funding need to be explained – e.g., I had no idea what was meant by "I-ASC". 7. The last paragraph of the introduction reads as a synopsis of the study, rather than aims – e.g., was the last sentence meant to describe a specific aim? It currently reads like results. 8. The "Survey development" includes information in addition to simply the development of the tool – participant recruitment and administration procedures. I suggest that this section be labelled Survey, with sub-headings of Development and Administration (or Procedures). Participants should be a separate section, with recruitment treated separately to description, with the latter left under Results. 9. No mention was made of ethical approval or consent procedures. 10. P. 5, line 57, a brief statement of the "aims of the wider research project" would be useful here. 11. P. 6, line 37, "D-efficient survey" requires explanation. 12. P. 9 – I would think the RCSLT would have data on the percentage of female: male professionals and cultural backgrounds, which may provide a better justification for believing the sample to be representative. 13. Can you provide the age range of participants? 14. The paragraph at the top of p. 10 was very difficult to follow – I suggest removing the long section from parentheses and place it in a block quote – this would also remove the need for nested parentheses. 15. Here and in other places, the shift to present tense was difficult to read and at times misleading. Present tense should be reserved for ongoing situations. I suggest that more conventional past tense language is used in describing findings – e.g., "there were no significant differences" rather than "there are no significant differences". This is a diffuse problem, and sometimes the tense shifts between present and past. 16. The discussion began with a statement about the feasibility of
--	---

	the DCEs as a research tool in AAC. I agree, but could not find this as a rationale for the study. I suggest it is either identified as such (and if so, this should be related to a better review of research), or moved to a section on research implications. 17. Also included in this first paragraph was that AAC professionals “found the tasks meaningful” – but how this was determined was not explained, nor were results reported. 18. This section is devoid of citations, an indication that there was no attempt to relate findings to previous research of relevance. This omission speaks to the lack of critical review of research in the introduction. 19. Research and clinical implications would have given readers a sense, respectively, of “where to next” and how the findings are relevant to clinical work. 20. I found the writing style of the discussion to be quite “clipped” – that is, sentences were difficult to follow because expression was so succinct, that I struggled to understand the meaning. Also, the authors fell into a habit of inappropriately abbreviating words – most notably “vocab” for “vocabulary.” Problems with expression were found in other parts of the Ms, but was at its most problematic here. Examples include the sentence starting on line 17, p. 8, and line 51, same page. There were numerous other examples. 21. Finally, the appendices were many and mostly quite large. My suggestion is that these be rationalised, although I did appreciate having access to the original tools, or sections of them. Still, I don’t think it necessary to include all items for the demographic section, nor to have them presented as in the survey. Depending on the journal’s policy, I wonder if some be supplementary files and the sample survey shortened so that sufficient information is provided to give the reader a sense of the task. Access to the full tool could be made available through contact with the corresponding author. I hope these comments are useful to the authors. Ultimately, I found much to recommend this work, but it needs to be better contextualised in terms of the international BMJ Open readership and previous research, and considerable attention paid to organisation and written expression.
--	---

REVIEWER	Jennifer Thistle Western Washington University, United States
REVIEW RETURNED	29-Apr-2019

GENERAL COMMENTS	Thank you for the opportunity to review the manuscript, "Finding the best fit: A discrete choice experiment on the decision making of augmentative and alternative communication professionals." The design and implementation is unique, interesting, and contributes to the field. Comments below progress sequentially through the manuscript. *On page 2, abstract, please spell out DCE in the Design as this is the first instance of it's use. *Throughout, consistent capitalization is needed when referring to "best-worst scaling". For instance, on page 3, no capitalization is used, later, some words are capitalized and others are not. Also, please provide background on what BWS-Case 1 is and rationale for using it here. *Throughout, review citation guidelines. Most citations are noted in superscript, yet in some places (e.g., page 4), citations are noted in brackets. *Page 4, Can I-ASC be spelled out? I realize it is part of the name of
---

	the study. Does style guidelines permit it being italicized? As an unfamiliar reader, I found this sentence awkward. Page 5, The first sentence of the Survey Development section is awkwardly worded. Consider re-wording. *Page 6, what does "how consistent the layout..." refer to? Consistent in what way or compared to what? *I am not familiar with the statistics described within and thus am not qualified to offer comments related to the analysis. *The results section nicely presented result only, without interpretation. The most important revisions are needed in the discussion section: *I would be interested in seeing more discussion connected back to existing research. In addition to how these results compare to the previous BWS Case 1 study; how does it compare to any other studies of clinical practice (e.g., Thistle & Wilkinson 2015) or research that that may not be put into practice yet (see Light et al., 2019 state of the science paper in the AAC journal). *A discussion of future directions could add to the utility of this manuscript. Again, thank you for the interesting and innovative study. Describing the decision-making of practitioners is critical to identify future research directions as well as bridge gaps between practice and research.
--	--

REVIEWER	Jennifer Cleland University of Aberdeen
REVIEW RETURNED	27-Jul-2019

GENERAL COMMENTS	I was surprised at the paucity of referencing. Given this is a DCE I would have expected a bit more background on the origins of the DCE methodology, and its translation into healthcare (see early work from Ryan). The fact this DCE has been informed by qualitative work needs to be foregrounded better - this is now best practice so reference to why it is so would make this paper stronger (e.g., Kløjgaard ME, Bech M, Søgaard R. Designing a stated choice experiment: the value of a qualitative process. Journal of Choice Modelling 2012;5:1–18.) A sentence or two explaining RUT would be good too - my main criticism is that this paper assumes the reader has knowledge of the approach, but that will not be the case in a general journal (which has admittedly published DCE studies, which you could refer to e.g., Scanlan et al 2018). Small numbers for a DCE - explain why this isn't an issue, or talk more about it being an issue in the discussion (if it is). I have to admit to finding this paper awfully dull. Is your point that a DCE is feasible or something more interesting? Your finding that AAC professionals' decision making can be strongly influenced by the characteristic of the child they are providing a system for is hugely reassuring but you don't really make much of this (beyond that statement). Could the paper be a bit more "human" in the introduction and discussion - how important is this in practice, how reassuring is it that professionals adjust their approach conditional on patient presentation (it seems a bit bleeding obvious to me, at the heart of the definition of professional but you could spell this out to make the paper more engaging).
--

VERSION 1 – AUTHOR RESPONSE

Reviewer 1

1. In the abstract, I suggest not using abbreviations without defining them – DCE being a case in point. Also, there were many incomplete sentences, including at lines 16, 21, 27.

We have written discrete choice experiment in full and revised the wording of the abstract to avoid incomplete sentences.

2. The language and expression suggested a very medical model focus, such as through referring to various types of disability as “conditions” and removing reference to the child when discussing the “vignettes” – e.g., p. 12, line 7, “a vignette with a receptive-expressive language gap,” and line 32 “a vignette motivated to communicate.” Similarly, I found the expression “a type of child seldom encountered” (p. 60) to be quite jarring. I suggest re-wording as “a child with a combination of characteristics seldom encountered.” Can I also suggest that people are “patients” (p. 9) only in health care settings. AAC decision-making does not always occur in a health setting, unless this is the case in the UK? I wonder if “consumer” could be substituted.

We thank the author for this feedback, as it is important to use person-centred language. We have revised the manuscript in several ways to help make it more person-centred, including removing references to “conditions” and deleting the phrase “a type of child seldom encountered”. We have kept the word “vignette”, as we believe it is important to emphasise that decisions are being made in response to a brief, four-line text, rather than a real child. However, we have altered the manuscript to use the phrase “child vignette” throughout. The only use of the word “patient” is in the sub-heading “Patient and public involvement”, which is required in the style guide of the journal. If the editor is in agreement, we are happy to change this to “Public involvement”.

3. As this journal has a wide readership, with many unfamiliar with AAC, a number of terms require definition – such as “pre-installed” vocabulary (although, if in relation to high technology, this is more usually referred to as “pre-programmed), “visual scene display,” “pragmatic organisation, “fixed or staged progression” (admittedly, although familiar with AAC literature, I don’t know what is meant by the latter).

We have endeavoured to make the manuscript as accessible as possible to readers without specialist knowledge of either AAC or DCEs, although this is sometimes difficult. We have provided explanations of the AAC terms used in Tables 1 and 2, and tried to aim these explanations at non-specialist readers. We have also included a reference to a standard work (Beukelman and Meranda, 2013), for readers who desire further information on AAC (page 6, paragraph 2).

4. The introduction would benefit from more information about previous relevant research. Citations are used to support statements, but this section should give the reader a sense of the state-of-the-art in relation to the topic of the study. I didn’t get this sense.

We have revised the introduction to give a sense of the current AAC landscape, and by previous findings which are relevant to the current study (introduction, paragraph 3; page 5, paragraph 3-4).

5. A stronger or perhaps more complete rationale is needed for the study. I can see great value in knowing how clinicians make decisions about AAC recommendations and in having a research tool that can be applied in contexts other than the UK. The tool and future research that explored the alignment of decisions and best practice recommendations (such as those of Beukelman and Meranda’s Participation Model, or clinical guidelines available through speech-language therapy/

pathology associations across the world) would certainly help in evaluation of practices. I wonder if the rationale could be strengthened, followed by specific aims (they currently are filtered through the latter part of the introduction) that better capture the focus on children and aided AAC. On the latter point, it was unclear if this was all aided AAC or only high technology, in light of the featuring of pre-installed vocabulary. Currently, the last paragraph of the introduction ends with addressing “both these aims” – but I struggled to know exactly what these were.

We have related the rationale to the above review of the current landscape, to show the gap in knowledge that the study aims to fill (page 5, paragraphs 4-5). We have also clarified that AAC attributes were intended to capture both high- and low-tech aided AAC systems, although some levels would be more relevant for high-tech systems (page 6, paragraph 4).

6. The study is contextualised as though it would be read by a UK audience only. Consideration needs to be given to the international readership: currently assumptions about who and how decisions are made about AAC are related to the UK only. For example, AAC decision-making does not always happen in teams. Also, reference to UK policies or funding need to be explained – e.g., I had no idea what was meant by “I-ASC”.

We have altered the text in various places to make it clearer for a worldwide audience (e.g. spelling out NHS). However, the study is specific to the UK context, thus we have amended the manuscript to make this limitation clear, and to make the reader more aware that different places may have other ways of working (page 2, paragraph 2). We have also clarified that I-ASC is the acronym for the wider research project (page 5, paragraph 2).

7. The last paragraph of the introduction reads as a synopsis of the study, rather than aims – e.g., was the last sentence meant to describe a specific aim? It currently reads like results.

This paragraph has been deleted for reasons of space.

8. The “Survey development” includes information in addition to simply the development of the tool – participant recruitment and administration procedures. I suggest that this section be labelled Survey, with sub-headings of Development and Administration (or Procedures). Participants should be a separate section, with recruitment treated separately to description, with the latter left under Results.

We have created separate sub-headings for survey development and survey administration.

9. No mention was made of ethical approval or consent procedures.

Information about ethical approval and consent procedures has been added (Survey administration, paragraph 1).

10. P. 5, line 57, a brief statement of the “aims of the wider research project” would be useful here.

The broad aims of the research project are summarised in the introduction (page 5, paragraph 2). The particular reference to the “aims of the wider research project” the reviewer mentions was non-essential and has been deleted for reasons of space.

11. P. 6, line 37, “D-efficient survey” requires explanation.

A sentence giving a rough, non-specialist description of D-efficiency has been added, together with a reference to a rigorous definition (page 7, paragraph 3).

12. P. 9 – I would think the RCSLT would have data on the percentage of female: male professionals and cultural backgrounds, which may provide a better justification for believing the sample to be representative.

We thank the reviewer for the suggestion. The RCSLT has provided some information which we have included as justification for the sample being roughly representative (page 10, paragraph 1).

13. Can you provide the age range of participants?

The age range has been added (page 10, paragraph 1).

14. The paragraph at the top of p. 10 was very difficult to follow – I suggest removing the long section from parentheses and place it in a block quote – this would also remove the need for nested parentheses.

We have revised the formatting of that paragraph to make it clearer (page 10, paragraph 4).

15. Here and in other places, the shift to present tense was difficult to read and at times misleading. Present tense should be reserved for ongoing situations. I suggest that more conventional past tense language is used in describing findings – e.g., “there were no significant differences” rather than “there are no significant differences”. This is a diffuse problem, and sometimes the tense shifts between present and past.

The manuscript has been changed so that findings are in the past tense throughout.

16. The discussion began with a statement about the feasibility of the DCEs as a research tool in AAC. I agree, but could not find this as a rationale for the study. I suggest it is either identified as such (and if so, this should be related to a better review of research), or moved to a section on research implications.

Assessing the feasibility of conducting a DCE in AAC has been explicitly stated in the introduction as one of the goals of the study (page 5, paragraph 5). The discussion of feasibility has been moved to later in the discussion section in order to emphasise other findings, and details have been added to show why the feasibility of a DCE in this area is noteworthy (page 15, final paragraph).

17. Also included in this first paragraph was that AAC professionals “found the tasks meaningful” – but how this was determined was not explained, nor were results reported.

This sentence has been removed and a more nuanced discussion of the acceptability of the DCE tasks added (page 15, final paragraph).

18. This section is devoid of citations, an indication that there was no attempt to relate findings to previous research of relevance. This omission speaks to the lack of critical review of research in the introduction.

Throughout the discussion, the findings have been compared and contrasted with the previous literature, with appropriate citations. This includes identifying the new contributions this manuscript makes to the literature.

19. Research and clinical implications would have given readers a sense, respectively, of “where to next” and how the findings are relevant to clinical work.

The conclusion has been revised to add how the findings of this study, together with the findings of the wider I-ASC project, have been used to create freely available resources for use in clinical practice. We have provided a link to the resources for readers who may find them useful.

20. I found the writing style of the discussion to be quite “clipped” – that is, sentences were difficult to follow because expression was so succinct, that I struggled to understand the meaning. Also, the authors fell into a habit of inappropriately abbreviating words – most notably “vocab” for “vocabulary.” Problems with expression were found in other parts of the Ms, but was at its most problematic here. Examples include the sentence starting on line 17, p. 8, and line 51, same page. There were numerous other examples.

We have revised the manuscript to attempt to make it clearer to the reader, including not abbreviating “vocab”. As part of this process, we have also re-structured the results section, including adding sub-headings, to make it easier to follow.

21. Finally, the appendices were many and mostly quite large. My suggestion is that these be rationalised, although I did appreciate having access to the original tools, or sections of them. Still, I don't think it necessary to include all items for the demographic section, nor to have them presented as in the survey. Depending on the journal's policy, I wonder if some be supplementary files and the sample survey shortened so that sufficient information is provided to give the reader a sense of the task. Access to the full tool could be made available through contact with the corresponding author.

We respectfully disagree with the reviewer, as we believe that each appendix provides important information to many readers. As they will be available only as supplementary online materials, we don't believe they would pose a burden on readers who do not find them of interest. Our justifications for including each appendix are:-

Appendix A – Selecting DCE attributes from amongst the BWS attributes is a crucial step in the construction of the survey. Thus, while the full list of potential attributes is not included in the main body of the paper for reasons of space, we believe it is important to make them easily accessible.

Appendix B – We believe it is important to be as transparent as possible about how data was collected, and the best way to do this is provide the survey instrument in full. Thus the interested reader is informed precisely what participants were asked, how, and in what order.

Appendix C – The details of the model selection process are not included in the main body of the text for reasons of space, and as they would not be informative to general readers without a background in econometrics/choice modelling. However, we believe it is still important to be transparent about the full details of the process, and hence have included them as an appendix.

Appendix D – Relative Interaction Attribute Importance (RIAI) is a novel measure, thus a formal definition is necessary to include, as there is no published work to reference instead. However, as the journal has a general readership, we believe it is best to include this as an appendix, rather than in the main body of the manuscript.

Reviewer 2

1. On page 2, abstract, please spell out DCE in the Design as this is the first instance of it's use.

We have spelled out DCE.

2. Throughout, consistent capitalization is needed when referring to "best-worst scaling". For instance, on page 3, no capitalization is used, later, some words are capitalized and others are not. Also, please provide background on what BWS-Case 1 is and rationale for using it here.

We have changed the capitalisation to be consistent. We have also expanded on the purpose of the

BWS survey, the rationale behind using this method, and the relationship between the earlier results and the DCE (page 5, paragraphs 4-5).

3. Throughout, review citation guidelines. Most citations are noted in superscript, yet in some places (e.g., page 4), citations are noted in brackets.

We have ensured all citations are correctly formatted.

4. Page 4, Can I-ASC be spelled out? I realize it is part of the name of the study. Does style guidelines permit it being italicized? As an unfamiliar reader, I found this sentence awkward.

We have re-written this sentence to make it clearer and italicised the name of the study (page 5, paragraph 2).

5. Page 5, The first sentence of the Survey Development section is awkwardly worded. Consider re-wording.

This sentence has been re-worded to make it clearer (page 6, paragraph 1).

6. Page 6, what does "how consistent the layout..." refer to? Consistent in what way or compared to what?

More detail has been added to clarify what this means (page 6, final paragraph), see also Table 2 for a non-specialist description.

7. I would be interested in seeing more discussion connected back to existing research. In addition to how these results compare to the previous BWS Case 1 study; how does it compare to any other studies of clinical practice (e.g., Thistle & Wilkinson 2015) or research that that may not be put into practice yet (see Light et al., 2019 state of the science paper in the AAC journal).

We have followed the reviewer's suggestion and compared and contrasted our results with previous findings in the literature throughout the discussion section. This includes the BWS case 1 study (page 15, paragraphs 3-4) and Thistle and Wilkinson (2015) (page 13, paragraph 4).

8. A discussion of future directions could add to the utility of this manuscript.

The conclusion has been amended to add that the findings from the I-ASC project, including this study, have been used to create resources for future use in clinical practice.

Reviewer 3

1. I was surprised at the paucity of referencing. Given this is a DCE I would have expected a bit more background on the origins of the DCE methodology, and its translation into healthcare (see early work from Ryan).

We have included references to systematic reviews of DCEs in healthcare, as well as a reference to the early translation work of Ryan (page 4, final paragraph).

2. The fact this DCE has been informed by qualitative work needs to be foregrounded better - this is now best practice so reference to why it is so would make this paper stronger (e.g., Kløjgaard ME, Bech M, Søgaard R. Designing a stated choice experiment: the value of a qualitative process. *Journal of Choice Modelling* 2012;5:1-18.)

We thank the reviewer for the suggestions, and have expanded the description of the attribute generation process, as well as adding a rationale as to why it is good practice, and included references (page 6, paragraph 1).

3. A sentence or two explaining RUT would be good too - my main criticism is that this paper assumes the reader has knowledge of the approach, but that will not be the case in a general journal (which has admittedly published DCE studies, which you could refer to e.g., Scanlan et al 2018).

We have added a brief description of the RUT approach for the general reader, and included a standard reference for readers who wish to learn more (page 8, paragraph 2).

4. Small numbers for a DCE - explain why this isn't an issue, or talk more about it being an issue in the discussion (if it is).

We have mentioned the sample size and discussed it, including putting it into context with other DCE studies in healthcare as well as the size of the AAC community in the UK (page 16, paragraph 3).

5. Is your point that a DCE is feasible or something more interesting?

We have added more detail to clarify the challenges of the project and why the feasibility of running a DCE in this area is noteworthy. However, we have moved this to later in the discussion section, to emphasise that it was not the central finding of the study (page 15, final paragraph). We have explicitly stated in the introduction that establishing the feasibility of running a DCE in AAC was an aim of the study (page 5, paragraph 5).

6. Your finding that AAC professionals' decision making can be strongly influenced by the characteristic of the child they are providing a system for is hugely reassuring but you don't really make much of this (beyond that statement). Could the paper be a bit more "human" in the introduction and discussion - how important is this in practice, how reassuring is it that professionals adjust their approach conditional on patient presentation (it seems a bit bleeding obvious to me, at the heart of the definition of professional but you could spell this out to make the paper more engaging).

It is true that AAC professionals reacting to characteristics of the child is perhaps not surprising, and have altered the discussion to reflect that (discussion paragraph 1). We have throughout the discussion emphasised that it is the magnitude of preference changes, and what sort of interactions are observed that are of interest. We have also related these to previous findings and recommendations in the AAC literature throughout the discussion.

VERSION 2 – REVIEW

REVIEWER	Teresa Iacono La Trobe University Australia
REVIEW RETURNED	19-Oct-2019
GENERAL COMMENTS	Thank you for the opportunity to review the revised version of this Manuscript (Ms). I found the organisation and overall quality of the Ms much improved, with most of the concerns noted in the previous review addressed. There remain some minor language and style issues, which need to be addressed. 1. The language still sometimes reflects a very medical model, starting with the first paragraph of the abstract. I don't see CP or autism as ongoing medical "reasons" – so suggest rewording this

	sentence to “Many children with varied disabilities can benefit from ...AAC.” On p. 3, line 20, suggest that “struggle” not be used, but rather a statement made that many children lack speech to meet functional needs. (“disorder” should come after “autism spectrum”). And rather than “within diagnoses” suggest, “within disability types”. 2. Tense shifts continue to be a problem – e.g., the first dot point in Strengths and Limitations is present tense, and the others are past tense. P. 4, line 49, suggest changing to “...there have been studies in which some important factors have been highlighted,” line 56, “addressed” rather than “addresses. There are many examples throughout – it may be that professional editing will help pick these up. 3. Use of “This” at the beginning of sentences, where the referent is unclear – e.g., p. 12 line 44 – what exactly was unexpected? There are other examples. 4. P. 8, line 5, suggest changing “Those who indicated they did not answered only demographic questions” to “Those who indicated they did not progressed directly to demographic questions that were at the end of the survey.” Then delete the last sentence of the para as it becomes redundant. 5. “e.g.,” is used throughout, where it should be written out (ie outside parentheses). 6. P. 10, line 12 – “experience on average of AAC” is poorly worded, suggest changing to “...they had, on average, 10 years of experience in AAC.” 7. P. 10, sentence beginning “The “constant” terms ...” is too long, so suggest changing to 2 sentences. Also I am not sure why constant is in quotation marks – this occurs for “ambitious” on p. Note that quotation marks should not be used for emphasis. 8. P. 11-12 – the subheadings here are not in a consistent format/style. I hope these comments are helpful, and congratulate the authors on an important piece of work.
--	---

VERSION 2 – AUTHOR RESPONSE

Reviewer

1. The language still sometimes reflects a very medical model, starting with the first paragraph of the abstract. I don't see CP or autism as ongoing medical “reasons” – so suggest rewording this sentence to “Many children with varied disabilities can benefit from ...AAC.” On p. 3, line 20, suggest that “struggle” not be used, but rather a statement made that many children lack speech to meet functional needs. (“disorder” should come after “autism spectrum”). And rather than “within diagnoses” suggest, “within disability types”.

We have adopted the suggested wording in the abstract as well as in the first paragraph of the introduction.

2. Tense shifts continue to be a problem – e.g., the first dot point in Strengths and Limitations is present tense, and the others are past tense. P. 4, line 49, suggest changing to “...there have been studies in which some important factors have been highlighted,” line 56, “addressed” rather than “addresses. There are many examples throughout – it may be that professional editing will help pick these up.

We have reviewed which tense is used throughout the manuscript.

3. Use of “This” at the beginning of sentences, where the referent is unclear – e.g., p. 12 line 44 – what exactly was unexpected? There are other examples.

We have reviewed the use of this language and changed it to clarify several sentences, including the example highlighted.

4. P. 8, line 5, suggest changing “Those who indicated they did not answered only demographic questions” to “Those who indicated they did not progressed directly to demographic questions that were at the end of the survey.” Then delete the last sentence of the para as it becomes redundant.

We have adopted the proposed wording

5. “e.g.,” is used throughout, where it should be written out (ie outside parentheses).

We have ensured “e.g.” is outside parentheses throughout the manuscript.

6. P. 10, line 12 – “experience on average of AAC” is poorly worded, suggest changing to “...they had, on average, 10 years of experience in AAC.”

We have adopted the proposed wording.

7. P. 10, sentence beginning “The “constant” terms ...” is too long, so suggest changing to 2 sentences. Also I am not sure why constant is in quotation marks – this occurs for “ambitious” on p. Note that quotation marks should not be used for emphasis.

We have removed the inverted commas from both instances, and divided up the relevant sentence to provide clarity.

8. P. 11-12 – the subheadings here are not in a consistent format/ style.

We have changed the formatting to bring the headings on these pages in line with the rest of the paper.